# Notational Analysis of Wheelchair Paralympic Table Tennis Matches

**DOI:** 10.3390/ijerph20053779

**Published:** 2023-02-21

**Authors:** Alessandro Guarnieri, Valentina Presta, Giuliana Gobbi, Ileana Ramazzina, Giancarlo Condello, Ivan Malagoli Lanzoni

**Affiliations:** 1Department of Medicine and Surgery, University of Parma, 43126 Parma, Italy; 2Department for Life Quality Studies, University of Bologna, 40136 Bologna, Italy

**Keywords:** performance analysis, performance indicators, match analysis, elite table tennis player, physical impairment

## Abstract

Paralympic table tennis is the third largest paralympic sport for the number of players. Performance analysis was conducted for the rally duration and interval and impact of serve, whilst none investigated the shots distribution among classes of physical impairment. Therefore, the purpose of this study was to conduct a notational analysis of international competitions in relation to the wheelchair classes. Five matches for each wheelchair class (C1-to-C5) were evaluated from 20 elite male right-handed players. Both players for each match were analyzed for the following performance indicators: strokes type, the area of ball bouncing, and the shots outcome. Backhand shots were the most used technique for all classes. The most used strokes for C1 players were backhand and forehand drive and backhand lob, while for C5 players they were backhand and forehand push and backhand topspin. Similar shots distribution was registered for C2-to-C5 players. The central and far-from-the-net zone was mainly reached by the serve for all classes. Errors shots were similar in all classes, whilst winning shots were more frequent in C1. The current notational analysis provided a meaningful performance modelling of indicators for coaches and athletes that can be used to design training programs for each class.

## 1. Introduction

Table tennis is one of the most popular sports in the world, included in the Olympic Games since 1988 [1]. Similarly, Paralympic table tennis (PTT) is the third largest Paralympic sport for the number of athletes, and it is practiced in more than 100 countries [2]. PTT players are divided into 11 classes for the competitions, according to their capabilities. Wheelchair players compete within class 1–5 (C1–C5), standing players within class 6–10, players with intellectual impairments in class 11. Players in C1 and C2 have the worse physical impairment, including tetraplegic players, whilst C5 wheelchair players are those with the best physical capabilities [3].

PTT performance was previously investigated for the epidemiology of sport injuries [4], with a special focus on the shoulder injury [5]. From a biomechanical perspective, the different techniques of the upper limb kinematics [6,7] and the trunk rotation [8] between able bodied and para table tennis players have been evaluated. Furthermore, the effects of psychological skills training [9] and the relationship between level of eye–hand coordination and performance outcome [10] have been detected in para-athletes. Players with intellectual impairments were investigated for their cognitive profile [11], their technical and tactical proficiency [12,13], their ability to adapt service and return [14], and some cognitive predictors of the performance [15].

For the implementation of performance modelling, the notational analysis is the process to collect information about the technical and tactical skills and the playing actions during a game [16]. Its purpose is to determine the most occurring performance indicators for a better design of training program and to optimize the performance [17]. Although performance indicators were largely determined for racket sports [18,19,20], several investigations are specifically available for table tennis [21,22,23,24], even if mainly focused on able-bodies athletes. Conversely, elite PTT game characteristics were marginally investigated for rally duration and interval and impact of serve [25,26]. Fuchs and colleagues [25] analyzed 227 PTT matches and found shorter rallies and a high direct impact of the serve for all female sitting players. Same results were also found for male sitting and standing para players with the highest level of impairments. Moreover, Da Silva and Reina [26] selected eight matches of the 2018 Paralympic Games. They registered shorter duration of rallies and longer rest time in wheelchair players with less impairments. Previous literature suggested the most important performance indicators in table tennis [21] including strokes type, impact position of the ball on the table, outcome of the action, and many others (steps, equipment, rally time). Strokes type was considered a technical performance indicator, whilst area of ball bouncing and shots outcome were classified as tactical performance indicators [21]. However, the characterization of shot distribution and area of ball bouncing is still lacking. Therefore, the purpose of this study was to conduct a notational analysis on wheelchair PTT matches for the determination of the strokes type, the area of ball bouncing, and the shots outcome according to the different wheelchair classes. A different distribution for the investigated variables among wheelchair classes is hypothesized, in light of the different levels of impairment and capabilities.

## 2. Materials and Methods

### 2.1. Sample

Twenty-five PTT matches were selected from international competitions between 2012 and 2018 (2012 London and 2016 Rio de Janeiro Olympic Games, 2015 European Championship, and 2018 World Championship). Therefore, 5 matches for each wheelchair class (C1–C5) and a total sample of 50 male elite players, ranked between 1st and 18th position at the time of the competition, were included in the analysis. All the selected players were right-handed, in order to totally exclude handedness influence on the selected performance indicators. The handedness of players was established according to which hand was used to hold the racket [27]. Thirty-eight players were European, 10 players were Asian, 1 player was South American, and 1 player was Oceanian. Every player used shake-hand grip, most of them competed with the classic inverted rubbers (pimples in) on their racket. Players with different types of rubbers were distributed as follows: 1 player with short pimple rubber on backhand side in C1; 3 players with long pimples rubber on backhand side and 1 player with short pimples on forehand side in C2; 4 players with long pimples on backhand side and 2 players with short pimples on backhand side in C3; 1 player with long pimples on backhand side in C4; 1 player with short pimples on backhand side in C5.

### 2.2. Procedures

The video of each match was freely available on the platform YouTube. The videos were downloaded and subsequently analyzed with Kinovea software (v. 0.9.4, Kinovea, Bordeaux, France) at a frequency of 10 frames per second. The camera was placed at a central and elevated position behind a player. An experienced table tennis coach, whilst watching the videos, collected the performance indicators of interest on a spreadsheet. Full data collection needed fifty hours, approximately.

For both players of each match, the strokes type, the area of ball bouncing, and the shots outcome were assessed for each shot. For strokes type, Table 1 presents the considered categories [28]:

Each shot was further classified as a backhand (BH) and forehand (FH) execution.

The area of ball bouncing was the area where the ball bounced after a player’s stroke. According to previous studies [1,29], the area of the table was divided into 6 equal rectangles and numerated from 1 to 6 (Figure 1).

The three categories of shots outcomes are presented in Table 2.

### 2.3. Analysis of Reliability

A single match was analyzed by three national table tennis coaches for the measurement of intra-operator reliability, whilst a single match was analyzed 3 times from the author (A.G.) who conducted the entire data analysis, for the measurement of inter-operator reliability. Therefore, Krippendorff’s alpha [30] was applied to calculate intra- and inter-operator reliability. For strokes type, area of ball bouncing and shots outcome, alpha equaled 0.99, 0.99, 1.00 for the intra-operator reliability, and 0.89, 0.94, 0.98 for the inter-operator reliability, respectively.

### 2.4. Statistical Analysis

Data were analyzed using the Statistical Package for the Social Sciences, version 25.0 (SPSS Inc., Chicago, IL, USA). The level of statistical significance was set at *p* < 0.05 for all computations. The normality assumption for each variable was verified using the Shapiro–Wilk test, which confirmed the normal distribution of data. One-way ANOVA was applied to ascertain the main effect of class on the investigated variables. Effect size for main effect was calculated as partial eta squared (*η_p_*^2^) and interpreted as small (0.01–0.06), medium (0.06 < *η_p_*^2^ < 0.14), and large effects (>0.14) [31]. In case of a significant main effect, Bonferroni post hoc was applied for multiple comparisons. Cohen’s d effect sizes (d) for each comparison were calculated and interpreted as trivial (<0.19), small (0.20–0.59), moderate (0.60–1.19), large (1.20–1.99), very large (2.00–4.00), and extremely large (>4.0) effects [32].

## 3. Results

A total of 1700 points (C1 = 328 pt., C2 = 393 pt., C3 = 354 pt., C4 = 291 pt., C5 = 334 pt.) and 8506 shots (C1 = 1111, C2 = 1687, C3 = 2366, C4 = 1658, C5 = 1683) were registered. The mean number of shots per rally in C1, C2, C3, C4, C5 classes were 3.4 ± 0.3, 4.3 ± 0.7, 6.3 ± 2.2, 5.7 ± 1.0, 5.0 ± 0.6, respectively.

### 3.1. Analysis of Strokes Type

Differences did not emerge for backhand (F = 0.050, *p* = 0.095) and forehand (F = 0.050, *p* = 0.095) shots distribution (Table 3).

A significant main effect emerged for D (F = 16.877, *p* < 0.001, *η_p_*^2^ = 0.600), L (F = 12.833, *p* < 0.001, *η_p_*^2^ = 0.533), P (F = 12.202, *p* < 0.001, *η_p_*^2^ = 0.520), T (F = 6.838, *p* < 0.001, *η_p_*^2^ = 0.378), and TCT (F = 5.579, *p* = 0.001, *η_p_*^2^ = 0.332). Post hoc comparisons among classes are shown in Table 4.

Considering the strokes-type distribution for each backhand and forehand execution and excluding serve, a significant main effect emerged for BH B (F = 2.651, *p* = 0.045, *η_p_*^2^ = 0.191), BH D (F = 9.045, *p* < 0.001, *η_p_*^2^ = 0.446), BH L (F = 15.283, *p* < 0.001, *η_p_*^2^ = 0.576), BH P (F = 4.534, *p* = 0.004, *η_p_*^2^ = 0.287), BH T (F = 9.606, *p* < 0.001, *η_p_*^2^ = 0.461), FH D (F = 6.152, *p* < 0.001, *η_p_*^2^ = 0.354), and FH P (F = 2.971, *p* = 0.029, *η_p_*^2^ = 0.209). Post hoc comparisons among classes are shown in Table 5.

No differences were obtained for backhand (F = 0.503, *p* = 0.734) and forehand (F = 0.503, *p* = 0.734) serves distribution (Table 6).

### 3.2. Analysis of Area of Ball Bouncing

The analysis of area of ball bouncing demonstrated a significant main effect for A1 (F = 3.870, *p* = 0.009, *η_p_*^2^ = 0.256), A2 (F = 2.650, *p* = 0.045, *η_p_*^2^ = 0.191), A6 (F = 7.262, *p* < 0.001, *η_p_*^2^ = 0.392), and OUT (F = 2.882, *p* = 0.033, *η_p_*^2^ = 0.204). Post hoc comparisons among classes are shown in Table 7.

### 3.3. Analysis of Shots Outcome

The analysis of shots outcome showed a significant main effect for Neutral shots (F = 11.328, *p* < 0.001, *η_p_*^2^ = 0.502), Winners (F = 7.842, *p* < 0.001, *η_p_*^2^ = 0.411) and Errors (F = 3.170, *p* = 0.022, *η_p_*^2^ = 0.220). Post hoc comparisons among classes are shown in Table 8.

The strokes-type distribution for error outcome demonstrated a significant main effect for BH B (F = 3.502, *p* = 0.014, *η_p_*^2^ = 0.237), BH D (F = 7.102, *p* < 0.001, *η_p_*^2^ = 0.387), BH L (F = 9.163, *p* < 0.001, *η_p_*^2^ = 0.449), BH T (F = 8.506, *p* < 0.001, *η_p_*^2^ = 0.431), FH D (F = 4.214, *p* = 0.006, *η_p_*^2^ = 0.272), FH P (F = 3.418, *p* = 0.015, *η_p_*^2^ = 0.236), FH T (F = 2.817, *p* = 0.036, *η_p_*^2^ = 0.200). Post hoc comparisons among classes are shown in Table 9.

The strokes-type distribution for winner outcome showed a significant main effect for BH L (F = 9.820, *p* < 0.001, *η_p_*^2^ = 0.466) and FH T (F = 4.587, *p* < 0.001, *η_p_*^2^ = 0.290). Post hoc comparisons among classes are shown in Table 10.

## 4. Discussion

The present study analyzed the most relevant shots characteristics during elite PTT matches, with the aim to find technical and tactical differences within wheelchair class of impairments. For that purpose, the strokes type, the area of ball bouncing, and the shots outcome were selected as performance indicators in this notational analysis. Previous literature about PTT match analysis was mainly based on the duration of the rallies and impact of the serve [25]. Moreover, duration of the rallies and rest time were analyzed in wheelchair players [26]. To the best of our knowledge, the present investigation should be considered innovative, giving remarkable achievements and useful findings.

Backhand shot is the most used technique by all the matches analyzed, with no significant differences among classes. This result is in contrast with those found by Malagoli Lanzoni et al. [1] during able-bodied table tennis matches. Indeed, forehand shot is preferred in able-bodied table tennis players in respect to backhand shots. Differently, PTT players used to play backhand stroke because it is considered easier to perform in front of the table. It is also mainly due to the fixed position of the wheelchair in the central zone of the table. In addition, less physical effort is required in performing backhand techniques. Conversely, forehand shot is preferred by able-bodied players in modern table tennis because of its effectiveness and winning outcome [1].

Serve in the strokes type was not considered for the current notational analysis to exclude its influence on the comparison since every rally starts with the serve. Push is the most used technique by all classes from C2 to C5. It is the shot more associated with a conservative playing style to return the opponents’ serves performed with a back-spin. Conversely, players in C1 demonstrated a lower percentage for push shots probably due to the different kind of serves with no-spin they received. Accordingly, the drive shot is the most used technique by C1 players because it is easier to play with no spin and high speed, resulting in its effectiveness in the categories with the worse physical impairment. Significant differences were also detected for lob and topspin shots. Lob stroke is especially used in C1, where the close-to-the-net shot is very difficult to be returned by players with worse physical impairment. Instead, the offensive shot topspin is largely played in C5 than in C1 and C2. It is probably linked to the high physical effort required in order to perform this offensive shot, which is the most important technique used in modern table tennis [1].

Considering the distribution of strokes type and including backhand and forehand executions, several evidence was derived. C1 players prefer to use backhand drive, forehand drive, and backhand lob. The two backhand shots are mainly induced by the fixed central position of the wheelchair. Classes from C2 to C5 have a similar distribution of strokes-type distinguishing forehand and backhand executions, showing a similar technical–tactical playing style. Backhand push is clearly the most used shot for those categories, while backhand drive and forehand push are similarly used by C2–C5 classes with no significant differences. Moreover, C5 shows a specific high distribution of backhand topspin in comparison with C1, directly linked with an offensive playing style for C5.

Regarding the serve, the backhand and forehand execution and the area of ball bouncing were analyzed. The serve assumes a key role in racket sports because an effective shot allows the serving player to tactically direct the rally [1]. It is important to underline a critical rule in Paralympics with respect to Olympic table tennis. If the receiver is in a wheelchair due to a physical disability, the rally is a let (needs to be repeated) if the ball after touching the receiver’s court returns in the direction of the net, or comes to rest on the receiver’s court, or leaves the receiver’s court after touching it by either of the sidelines [33]. In the current study, most of the serves are played with the backhand technique, and no significant differences were detected among classes in comparing the backhand or forehand execution. This result shows a peculiar difference between wheelchair Paralympic and Olympic table tennis players. Indeed, high level table tennis able-bodied athletes showed the opposite serves’ distribution [1]. The position of the body with respect to the table is probably the main reason why players prefer one technique to the other. Considering the area of ball bouncing of the serves, most of them are directed to the central and far-from-the-net zone (area 6) for all classes, with some significant differences among them. The present investigation shows that all the wheelchair PTT prefer to play long serves in the far-from-the-net areas (area 1, 5, 6). This data contrast with those collected by Malagoli Lanzoni and colleagues [1] in able-bodied table tennis matches, who registered most of serves bouncing in close-to-the-net areas (area 2, 3, 4). These contrasting results could be the consequence of the different serves’ rules between the two categories, Paralympic and Olympic table tennis, respectively. In fact, a close-to-the-net serve can result in non-valid in wheelchair PTT. In addition, a short serve can allow the opponent to send the ball in very angled areas. On the contrary, able-bodied players often use close-to-the-net serve to avoid an immediate attack from the opponent [1].

Considering the outcome of the actions, neutral shot is the most frequent type, with significant differences in comparing C1 with all the other classes. Errors shot distribution is similar in all the wheelchair classes, except for C1 compared to C4. Instead, winning shot is more frequent in C1 than in C3, C4, and C5. This is probably due to the best physical capabilities of these players, able to reach more areas of the table. The result is also linked to the method used to collect this specific performance indicator. Indeed, the three types of outcomes were chosen because of their reliability in the data collection.

Several differences among classes were detected for strokes type considering the error or winner outcome. In C1, the backhand drive shot is the technique with the highest error outcome compared to all the other classes. It could be mainly due to the very high frequency of execution during the rallies. Forehand drive is also connected with the error outcome of the action for C1, because of the complexity in playing out-of-the-body shots for wheelchair PTT. Forehand drive is followed by backhand lob, which is one of the most characteristic shots in C1. Classes from 2 to 5 show a similar distribution about shots with an error outcome. Backhand drive and backhand push are the two strokes more linked with errors in those classes. Moreover, backhand topspin in C5 is more related with an error outcome. The backhand lob is the stroke with the best winning outcome in C1 and C2, with a significant difference compared with C3, C4, and C5. Therefore, backhand lob is linked with winner and error outcomes at the same time in C1. These results represent the importance of this technique that allows to win several rallies during the game, but it is also connected with a risky playing style. Forehand topspin is the most winning shot for C5, confirming the importance of this offensive technique for wheelchair PTT players who are less physically impaired, similar to Olympic players. Nevertheless, C4 and C5 players prefer to perform topspin with the backhand side, probably due to the fixed central position of the wheelchair which does not allow them to play many forehand shots. In addition, backhand drive is the second winning shot for both C1 and C5.

Practical implications for coaches and players can be driven by the current notational analysis. Considering the shots per rally distribution, C1 players need to play a lower number of shots during the training session compared to the other players. Fuchs et al., investigating the rally length in PTT matches, observed that rally length was shorter in C1 than other classes [25], as confirmed by the current notational analysis. Limited mobility for C1 players prevents them from playing long rallies. Therefore, for C1 players it is recommended to plan exercises with a limited number of shots per rally (four maximum). Conversely, for players in C2 to C5 classes it is recommended to plan technical–tactical exercises based on more than four shots/rally. The shots distribution suggests a different physical training session based on a backhand-oriented style of play for wheelchair Paralympic athletes. Backhand and forehand drive, and backhand lob need to be extensively performed during C1 training sessions. Conversely, C5 players should emphasize activities on backhand and forehand push, and backhand and forehand topspin. All the other categories show similar characteristics, and it suggests planning technical–tactical exercises without differences. Moreover, regarding serves, all the players need to train mainly on the long serve directed to the far-from-the-net areas.

In future, the research could be implemented comparing female, left-handed, ages, and non-elite players. Furthermore, standing Paralympic players could be considered for a future comparison with wheelchair and able-bodied table tennis players.

The present study has some limitations that need to be addressed and can serve as guidance for future research. Materials used by the participants could influence the technical and tactical variables. Indeed, the present investigation included players with long pimples rubbers, which is typically used by defenders. Therefore, rubber could have influenced the present data collection. However, it was not possible to totally exclude this kind of rubbers from the analysis, because long pimples are frequently used in PTT. Moreover, due the fact that the notational analysis was conducted on video freely available on YouTube, demographic characteristics of the players were not available. Height, weight, sitting height, upper limb length, and type of impairments could not be considered as potential confounding factors.

## 5. Conclusions

This study represents the first analysis of shots distribution in elite wheelchair table tennis players. In conclusion, several inter-class differences were found, specifically between the first (more impaired players) and the fifth class (less impaired players) and these findings can have a meaningful practical implication for players and coaches. Players, coaches, physical trainers, and performance analysts can use this information to plan specific training sessions and to improve the decision-making process during the game.

The understanding of the structure of the sport should be provided with detailed technical and tactical analyses in every discipline, in order to improve education and performance in both Olympic and Paralympic sports.

## Figures and Tables

**Figure 1 ijerph-20-03779-f001:**
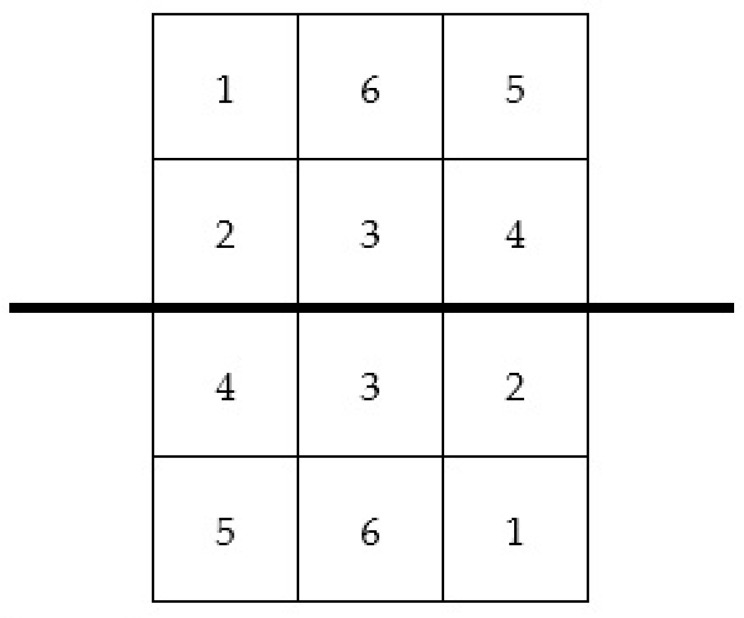
Six-area subdivision of each table tennis side. The thick line represents the net.

**Table 1 ijerph-20-03779-t001:** Description of strokes-type categories.

Strokes Type	Description
Block (B)	A defensive stroke played to response to a topspin in a passive way.
Drive (D)	An interlocutory stroke impressing no effect on the ball.
Flick (F)	An attacking stroke played when the ball bounces close to the net.
Lob (L)	A defensive stroke performed playing the ball high (in PTT, it became an attacking stroke impressing down-spin on the ball with the aim of preventing the opponent from hitting the ball).
Long pimples attack (LP)	An attacking stroke played using a special type of rubber (long pimples).
Push (P)	An interlocutory stroke impressing back-spin on the ball.
Serve (S)	The first shot of the rally, played impressing on the ball different spin (back-spin, top-spin or side-spin).
Smash (SM)	An attacking stroke impressing no effect on the ball played when it is high.
Topspin (T)	An attacking stroke impressing a top-spin effect on the ball.
Top-counter-top (TCT)	A counter-attacking stroke played to response to a topspin in an active way.

**Table 2 ijerph-20-03779-t002:** Description of shots outcome categories.

Shots Outcomes	Description
Neutral	A non-rally ending shot.
Error	A rally-ending shot after which the ball went outside or in the net; the type error (net/out) was also registered.
Winner	A rally-ending shot after which the ball was not touched by the opponent.

**Table 3 ijerph-20-03779-t003:** Distribution of backhand and forehand executions for each class (mean ± SD).

	C1	C2	C3	C4	C5
BH	67.3 ± 16.4%	65.3 ± 15.2%	66.4 ± 11.0%	66.9 ± 15.5%	67.9 ± 10.0%
FH	32.7 ± 16.4%	34.7 ± 15.2%	33.6 ± 11.0%	33.1 ± 15.5%	32.1 ± 10.0%

BH = backhand; C1–C5 = wheelchair class 1 to 5; FH = forehand.

**Table 4 ijerph-20-03779-t004:** Distribution of strokes type (serve excluded) for each class (mean ± SD).

	C1	C2	C3	C4	C5	*p* (d)
B	4.6 ± 5.2%	11.4 ± 12.3%	16.1 ± 9.4%	10.0 ± 8.2%	13.4 ± 7.3%	
D	59.4 ± 12.1%	32.1 ± 13.0%	20.1 ± 14.9%	28.8 ± 17.6%	15.2 ± 5.7%	C1 vs. C2: < 0.001 (2.17) C1 vs. C3: < 0.001 (2.90) C1 vs. C4: < 0.001 (2.03) C1 vs. C5: < 0.001 (4.67)
F	0	0	0.1 ± 0.2%	0.4 ± 1.1%	0.3 ± 0.6%	
L	15.3 ± 10.7%	6.0 ± 7.1%	0.4 ± 0.8%	0.3 ± 0.7%	0.2 ± 0.4%	C1 vs. C2: = 0.007 (1.02) C1 vs. C3: < 0.001 (1.96) C1 vs. C4: < 0.001 (1.98) C1 vs. C5: < 0.001 (1.99)
LP	0	5.9 ± 9.6%	4.3 ± 5.8%	2.9 ± 4.6%	0	
P	12.9 ± 7.3%	36.2 ± 13.8%	43.9 ± 12.8%	44.3 ± 18.9%	47.7 ± 7.3%	C1 vs. C2: = 0.002 (2.11) C1 vs. C3: < 0.001 (2.98) C1 vs. C4: < 0.001 (2.19) C1 vs. C5: < 0.001 (4.77)
SM	3.7 ± 3.6%	2.9 ± 2.5%	1.4 ± 1.4%	1.8 ± 1.3%	1.5 ± 1.2%	
T	4.1 ± 5.3%	5.5 ± 5.1%	13.6 ± 9.6%	11.6 ± 7.4%	20.5 ± 10.9%	C1 vs. C5: < 0.001 (1.91) C2 vs. C5: = 0.001 (1.76)
TCT	0	0	0.1 ± 0.2%	0.5 ± 0.9%	1.3 ± 1.3%	C1 vs. C5: = 0.003 (1.41)

B = block; C1–C5 = wheelchair class 1 to 5; D = drive; F = flick; L = lob; LP = long pimple attack; P = push; SM = smash; T = topspin; TCT = top-counter-top; d = Cohen’s d effect size.

**Table 5 ijerph-20-03779-t005:** Distribution of strokes type (serve excluded) in backhand and forehand executions for each class (mean ± SD).

	C1	C2	C3	C4	C5	*p* (d)
BH B	1.7 ± 2.3%	8.5 ± 11.4%	12.1 ± 8.5%	7.6 ± 6.9%	9.2 ± 4.5%	C1 vs. C3: = 0.030 (1.67)
BH D	36.1 ± 8.3%	18.7 ± 13.9%	13.7 ± 11.2%	15.6 ± 10.9%	12.1 ± 4.4%	C1 vs. C2: = 0.004 (1.52) C1 vs. C3: < 0.001 (2.72) C1 vs. C4: = 0.001 (2.12) C1 vs. C5: < 0.001 (3.61)
BH L	14.4 ± 9.5%	4.8 ± 5.8%	0.1 ± 0.3%	0.3 ± 0.7%	0	C1 vs. C2: = 0.001 (1.80) C1 vs. C3: < 0.001 (4.20) C1 vs. C4: < 0.001 (4.11) C1 vs. C5: < 0.001 (4.40)
BH P	10.6 ± 7.2%	23.7 ± 11.6%	30.7 ± 15.2%	31.9 ± 22.0%	33.3 ± 8.2%	C1 vs. C3: = 0.024 (1.69) C1 vs. C4: = 0.013 (1.30) C1 vs. C5: = 0.007 (2.94)
BH T	1.0 ± 2.0%	1.7 ± 2.4%	6.4 ± 6.0%	5.9 ± 3.9%	12.1 ± 6.4%	C1 vs. C5: < 0.001 (2.34) C2 vs. C5: < 0.001 (2.26) C4 vs. C5: = 0.039 (0.92)
FH B	2.9 ± 4.1%	2.9 ± 3.0%	4.0 ± 3.8%	2.4 ± 1.7%	4.1 ± 3.0%	
FH D	23.4 ± 15.4%	13.4 ± 5.9%	6.4 ± 5.5%	12.5 ± 13.7%	3.0 ± 1.8%	C1 vs. C3: = 0.004 (1.47) C1 vs. C5: < 0.001 (1.86)
FH P	2.3 ± 2.3%	12.5 ± 9.2%	13.2 ± 11.4%	12.4 ± 11.3%	14.5 ± 7.8%	C1 vs. C5: = 0.042 (2.12)
FH T	3.0 ± 3.5%	3.8 ± 4.3%	7.2 ± 6.5%	5.7 ± 5.6%	8.4 ± 5.2%	
Others	4.6 ± 3.8%	9.9 ± 9.6%	6.2 ± 5.5%	5.6 ± 4.8%	3.2 ± 1.9%	

B = block; BH = backhand; C1–C5 = wheelchair class 1 to 5; D = drive; F = flick; FH = forehand; L = lob; LP = long pimple attack; P = push; SM = smash; T = topspin; TCT = top-counter-top; d = Cohen’s d effect size.

**Table 6 ijerph-20-03779-t006:** Distribution of backhand and forehand serve executions for each class (mean ± SD).

	C1	C2	C3	C4	C5
BH Serve	71.8 ± 29.2%	72.4 ± 28.8%	60.1 ± 33.3%	78.5 ± 17.7%	67.6 ± 38.7%
FH Serve	28.2 ± 29.2%	27.6 ± 28.8%	39.9 ± 33.3%	21.5 ± 17.7%	32.4 ± 38.7%

BH = backhand; C1–C5 = wheelchair class 1 to 5; FH = forehand.

**Table 7 ijerph-20-03779-t007:** Distribution of area of ball bouncing after serve for each class (mean ± SD).

	C1	C2	C3	C4	C5	*p* (d)
A1	26.0 ± 14.1%	27.2 ± 22.9%	9.9 ± 8.3%	11.8 ± 9.4%	10.8 ± 9.6%	
A2	8.8 ± 9.3%	4.3 ± 4.3%	2.3 ± 3.7%	1.5 ± 3.1%	4.6 ± 4.9%	C1 vs. C4: = 0.049 (1.05)
A3	6.3 ± 7.6%	9.4 ± 8.2%	10.5 ± 10.2%	11.0 ± 7.2%	7.9 ± 6.6%	
A4	0.9 ± 1.5%	1.3 ± 1.9%	0.6 ± 1.4%	0.4 ± 1.4%	0	
A5	24.6 ± 14.7%	21.6 ± 14.5%	20.4 ± 12.4%	25.7 ± 20.1%	11.1 ± 9.1%	
A6	28.2 ± 14.1%	32.9 ± 16.3%	53.7 ± 14.8%	47.6 ± 22.3%	64.4 ± 18.7%	C1 vs. C3: = 0.021 (1.76) C1 vs. C5: < 0.001 (2.19) C2 vs. C5: = 0.002 (1.80)
OUT	3.1 ± 2.5%	2.0 ± 2.0%	1.2 ± 1.7%	0.6 ± 1.3%	0.9 ± 1.5%	C1 vs. C4: = 0.047 (1.25)
NET	2.1 ± 3.0%	0.5 ± 1.3%	1.2 ± 1.6%	1.4 ± 1.9%	0.3 ± 0.9%	
NS	0	0.7 ± 1.5%	0.2 ± 0.7%	0	0	

A1–A6 = Area of ball bouncing 1 to 5; C1–C5 = wheelchair class 1 to 5; NET = missed serve with the ball to the net; NS = Non-valid Serve (serve was repeated); OUT = missed serves with the ball out of the table; d = Cohen’s d effect size.

**Table 8 ijerph-20-03779-t008:** Distribution of shots outcome for each class (mean ± SD).

	C1	C2	C3	C4	C5	*p* (d)
Neutrals	70.9 ± 4.4%	76.4 ± 5.0%	82.7 ± 5.3%	82.2 ± 4.9%	79.9 ± 2.8%	C1 vs. C3: < 0.001 (2.42) C1 vs. C4: < 0.001 (2.42) C1 vs. C5: = 0.001 (2.44) C2 vs. C3: = 0.037 (1.22)
Errors	20.1 ± 3.9%	17.2 ± 4.0%	14.4 ± 5.3%	14.1 ± 4.8%	16.7 ± 3.6%	C1 vs. C4: = 0.032 (1.60)
Winners	8.9 ± 4.4%	6.4 ± 3.2%	2.9 ± 2.4%	3.8 ± 1.8%	3.4 ± 1.8%	C1 vs. C3: < 0.001 (1.69) C1 vs. C4: = 0.002 (1.52) C1 vs. C5: = 0.001 (1.64)

C1–C5 = wheelchair class 1 to 5; d = Cohen’s d effect size.

**Table 9 ijerph-20-03779-t009:** Distribution of strokes type in error outcome for each class (mean ± SD).

	C1	C2	C3	C4	C5	*p* (d)
BH B	2.9 ± 3.5%	10.7 ± 13.7%	17.4 ± 8.9%	14.8 ± 12.7%	18.3 ± 11.1%	C1 vs. C3: = 0.038 (2.14) C1 vs. C5: = 0.022 (1.87)
BH D	39.0 ± 15.2%	20.4 ± 15.9%	12.7 ± 9.5%	17.1 ± 12.4%	16.0 ± 5.7%	C1 vs. C2: = 0.016 (1.20) C1 vs. C3: < 0.001 (2.08) C1 vs. C4: = 0.003 (1.57) C1 vs. C5: = 0.001 (2.00)
BH L	12.3 ± 11.1%	5.0 ± 5.8%	0.3 ± 1.0%	0	0	C1 vs. C3: < 0.001 (1.52) C1 vs. C4: < 0.001 (1.57) C1 vs. C5: < 0.001 (1.57)
BH LP	0	5.0 ± 8.6%	3.5 ± 4.9%	1.2 ± 3.8%	0	
BH P	6.1 ± 7.2%	11.8 ± 7.1%	15.9 ± 9.8%	13.7 ± 10.2	13.5 ± 9.0%	
BH S	4.8 ± 5.1%	3.1 ± 3.7%	2.0 ± 2.3%	2.4 ± 2.8%	1.0 ± 1.7%	
BH T	0.5 ± 1.6%	0.7 ± 2.3%	7.7 ± 7.9%	8.2 ± 5.3%	12.5 ± 7.8%	C1 vs. C4: = 0.036 (1.97) C1 vs. C5: < 0.001 (2.13) C2 vs. C5: < 0.001 (2.05)
FH B	2.4 ± 3.4%	5.7 ± 4.6%	8.2 ± 5.8%	6.8 ± 9.0%	8.8 ± 6.1%	
FH D	23.3 ± 20.2%	15.9 ± 7.6%	8.7 ± 5.6%	11.3 ± 10.0%	4.6 ± 3.6%	C1 vs. C5: = 0.005 (1.29)
FH P	1.1 ± 2.4%	9.9 ± 7.9%	9.3 ± 6.9%	10.3 ± 7.0%	8.8 ± 6.9%	C1 vs. C2: = 0.040 (1.51) C1 vs. C4: = 0.027 (1.76)
FH T	1.7 ± 2.8%	5.1 ± 4.0%	11.2 ± 8.4%	10.7 ± 12.1%	10.7 ± 9.3%	
Others	6.0 ± 5.4%	6.7 ± 6.5%	3.1 ± 3.5%	3.3 ± 4.4%	5.8 ± 3.3%	

B = block; BH = backhand; C1–C5 = wheelchair class 1 to 5; D = drive; F = flick; FH = forehand; L = lob; LP = long pimple attack; P = push; SM = smash; T = topspin; TCT = top-counter-top; d = Cohen’s d effect size.

**Table 10 ijerph-20-03779-t010:** Distribution of strokes type in winner outcome for each class (mean ± SD).

	C1	C2	C3	C4	C5	*p* (d)
BH B	1.1 ± 3.5%	2.9 ± 4.8%	6.7 ± 13.1%	5.3 ± 11.7%	2.2 ± 4.9%	
BH D	27.9 ± 10.9%	13.6 ± 18.3%	7.4 ± 10.5%	13.9 ± 15.0%	20.8 ± 34.6%	
BH L	30.2 ± 18.3%	24.8 ± 27.3%	0	2.0 ± 6.3%	0	C1 vs. C3: < 0.001 (2.33) C1 vs. C4: = 0.001 (2.06) C1 vs. C5: < 0.001 (2.33) C2 vs. C3: = 0.006 (1.28) C2 vs. C4: = 0.014 (1.15) C2 vs. C5: = 0.006 (1.28)
BH P	5.4 ± 8.0%	3.2 ± 4.5%	23.3 ± 32.6%	14.8 ± 14.1%	8.9 ± 12.6%	
BH SM	9.2 ± 12.3%	1.3 ± 2.6%	1.0 ± 3.2%	4.6 ± 10.8%	1.7 ± 5.3%	
BH T	0.8 ± 2.6%	4.1 ± 9.5%	15.2 ± 31.0%	12.4 ± 14.1%	17.4 ± 14.9%	
FH B	0.6 ± 2.0%	9.0 ± 17.8%	1.0 ± 3.2%	9.5 ± 17.1%	2.9 ± 6.0%	
FH D	13.4 ± 9.1%	13.1 ± 12.4%	9.3 ± 22.3%	8.9 ± 13.5%	6.4 ± 16.0%	
FH LP	0	9.4 ± 15.4%	7.1 ± 13.2%	0	0	
FH P	2.9 ± 7.1%	3.4 ± 6.2%	6.3 ± 9.0%	6.1 ± 12.7%	6.5 ± 9.8%	
FH SM	4.6 ± 8.2%	9.7 ± 12.8%	9.6 ± 13.4%	7.0 ± 9.9%	6.1 ± 8.9%	
FH T	0.8 ± 2.4%	3.6 ± 5.1%	8.1 ± 9.3%	12.9 ± 18.7%	21.0 ± 15.3%	C1 vs. C5: < 0.001 (1.84) C2 vs. C5: = 0.002 (1.53) C3 vs. C5: = 0.019 (1.02)
Others	3.1 ± 7.2%	2.1 ± 4.7%	5.0 ± 15.8%	2.5 ± 5.6%	6.1 ± 11.2%	

B = block; BH = backhand; C1–C5 = wheelchair class 1 to 5; D = drive; F = flick; FH = forehand; L = lob; LP = long pimple attack; P = push; SM = smash; T = topspin; TCT = top-counter-top; d = Cohen’s d effect size.

## Data Availability

Not applicable.

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
