# Peer review of "Notational Analysis of Wheelchair Paralympic Table Tennis Matches"

_ijerph, 2023, doi:10.3390/ijerph20053779_

Round 1
Reviewer 1 Report
I think this kind of analysis and research are really important to improve understanding of demands and characteristics of any sport. Generally, it is a well-written paper. I have a couple of minor amendments requests below for authors
In abstract, presenting some important statistical values of the findings would be ideal.
In introduction, authors mentioned and cited (reference 22 and 23) that limited study investigated for rally duration and interval and impact of serve, some details of the methods and findings of those studies should be briefly given.
Material methods
Line 85-123 – I would suggest to present those categories in a table instead of bullet points, which is more appropriate regarding to academic writing style
It would be more ideal to present Cohen d effect sized as “d = …” instead of “ES”.
Line 322-324 – please add this part into the previous paragraph.
Reviewer 2 Report
This paper studied notational analysis of international competitions in relation to the wheelchair classes.Because there is Few studies investigated the shots distribution among classes of physical impairment, this paper proposes the notational analysis provided a meaningful performance modeling of indicators for coaches and athletes that can be used to design training programs for each class.
Major points:
1. The significance of this paper is due to the background of the characterization of shot distribution and area of ball bouncing is still lacking, this study was to conduct a notational analysis on wheelchair PPT matches for the determination of the strokes type, the area of ball bouncing, and the shots outcome according to the different wheelchair classes, including the hypothesis ,different distribution for the investigated variables, and the interpretations are very detailed and well-grounded. However the author did not summarize and review the research results related of the predecessors, so that the reader did not understand well the latest progress of the current domestic and foreign studies, and did not have a good exposition of the theoretical and matches practice. Please sort out the latest research results and let us understand clearly.
2. The sample selection was very careful, process and equipments are professional enough.It can also be understood by people in non-related professional fields. The focus of this article is Statistical analysis, undeniably, it is elaborated and analysis process is clear and direct. However we can do better in data presentation, for example, the visual presentation of elegant charts, for well explained, understandable and intelligible.
3. Through scientific and rigorous notational analysis, the strokes type, the area of ball bouncing, and the shots outcome were selected as performance indicators. And then explain the experimental, theoretical and practical basis of these the three one by one.
But the competition rules of each Paralympic Games and the physical conditions of the disabled must be taken into account. Therefore, it is still necessary to combine and consult previous researches and literatures, whether we have innovative and remarkable achievements and findings than others. The author can emphasize to make the conclusion more sufficient and well-knit.
4. In terms of conclusions, that is positive,is meaningful for us, but what are the implications for our current practical significance and subsequent academic research? I hope the author can conduct in-depth discussion and analysis to explore. In addition, the limitations of this paper are well stated, and what are the restrictive conditions and the parts that are well controlled in the experiment process explained to the readers, so as to help the subsequent improvement. There is insufficient interpretation and literature support for our results. For example,why hasn't this topic been discussed in previous studies? Is it due to conditions or for some reason? The prospect of the conclusion and the significance and importance of sports, the author could add explanations.
Reviewer 3 Report
Dear Authors.
The following is a review of the article entitled " Notational Analysis of Wheelchair Paralympic Table Tennis Matches”. The aim of this study was to conduct a notational analysis of international competitions in relation to the wheelchair classes. Thank you very much for thinking of me as a reviewer for this study.
After carefully reading the manuscript, I set forth comments and suggestions for the authors:
Abstract: Correct but improve the practical application of the results obtained. Validation of the instrument and the data should be mentioned. Some statistical data showing significance should be implemented.
Keywords: Added Observational methodology
Introduction: Line 28-35- This paragraph should have some reference to confirm this information.
Materials and Methods: Correct
Results: Correct. Too many tables. Reduce.
Discussion: Correct.
Conclusions: Added practical application.
References: Corrects.
Round 2
Reviewer 2 Report
I have no further comments.